# Rutin-Activated Nuclear Factor Erythroid 2-Related Factor 2 (Nrf2) Attenuates Corneal and Heart Damage in Mice

**DOI:** 10.3390/ph17111523

**Published:** 2024-11-12

**Authors:** Promise M. Emeka, Lorina I. Badger-Emeka, Krishnaraj Thirugnanasambantham, Abdulaziz S. Alatawi

**Affiliations:** 1Department of Pharmaceutical Science, College of Clinical Pharmacy, King Faisal University, Al Ahsa 31982, Saudi Arabia; 221400300@student.kfu.edu.sa; 2Department of Biomedical Science, College of Medicine, King Faisal University, Al Ahsa 31982, Saudi Arabia; lbadgeremeka@kfu.edu.sa; 3Pondicherry Centre for Biological Science and Educational Trust, Sundararaja Nagar, Puducherry 605004, India

**Keywords:** antioxidants, corneal injuries, isoproterenol, nuclear factor erythroid 2-related factor, oxidative stress, reactive oxygen species, rutin

## Abstract

**Background:** Corneal degeneration is a form of progressive cell death caused by multiple factors, such as diabetic retinopathy. It is the most well-known neural degenerative disease caused by macular degeneration in the aged and those with retinitis pigmentosa. Myocardial infarction is becoming a more common burden, causing cardiomyocyte degeneration, ischemia, and heart tissue death. This study examined the preventive effects of rutin on isoproterenol (ISO)-induced oxidative damage (that is, inflammation) on rabbit corneal epithelial cells and mouse heart injuries. **Methods:** These investigations involved a cytotoxicity test, biochemical analysis, qRT-PCR, Western blotting, and mouse cardiac histopathology. **Results:** The results showed that rutin enhanced ADH7 and ALDH1A1, retinoic acid signaling components in SIRC1 rabbit corneal cell lines. The production of NO by ocular epithelial cells was significantly reduced. It reduced cTnT and cTnI, CK-MB, and LDH contents in mouse cardiac tissue. The nuclear expressions of Nrf2, Sirt, and HO-1 were all increased by rutin. Docking studies revealed a good interaction between rutin and the Keap protein, enhancing Nrf2 nuclear activity. **Conclusions:** This showed that rutin can potentially enhance ADH7 and ALDH1A1 corneal signaling components, preventing corneal degeneration and mitigating ISO-induced myocardial infarction (MI) via Keap/Nrf2 expressions.

## 1. Introduction

The cornea is a target for both disease and environmental assaults as the transparent outermost membrane of the eye. It protects and aids vision while also filtering ultraviolet light [1,2]. Injuries can be due to aging and complications from diabetes mellitus, scarring different layers and leading to structural changes [3]. These changes are largely due to inflammation, which leads to oxidative stress that generates reactive oxygen species [4]. Cardiovascular diseases (CVDs) are the primary cause of global disability and death, and their burden has increased at an alarming rate [5]. Myocardial infarction (MI) is one of the most common CVDs and is a leading cause of death and disability worldwide, particularly in developing countries [6]. Efforts to reduce its incidence and improve treatment options are crucial in addressing this pressing public health issue. Studies indicate that oxidative stress-mediated inflammatory responses may be involved in MI [7,8].

ISO is a synthetic catecholamine and β-adrenergic agonist that has been found to cause oxidative stress in the tissues of the eye and myocardium [9,10]. ISO produces highly cytotoxic free radicals that cause cellular dysfunction, elevated lipid peroxidation, modified antioxidant enzyme activities, and (in the heart) necrosis that resembles an infarct [11].

Nuclear factor erythroid 2-related factor 2 (Nrf2), a crucial transcription factor, is known to trigger the production of endogenous antioxidant enzymes in response to oxidative stress [12]. Under homeostatic circumstances, Nrf2 binds to Kelch-like ECH-associated protein 1 (Keap1) and resides inside the cytoplasm in an inactive form. ROS cause Nrf2 to migrate into the nucleus after release from the regulatory Keap1-Nrf2 complex. Previous studies indicate that Nrf2 is activated in injured corneal epithelia and plays a protective role in wound healing [13]. Previously, Ma [14] reported that Nrf2 was essential for stimulating endogenous antioxidant enzymes, which guard against oxidative damage in several illnesses. Hence, antioxidant enzymes play an important role in detoxifying generated reactive oxygen species (ROS) [15].

Flavonoids are polyphenolic compounds found in plants which play an essential role in detoxifying free radicals. Rutin is a flavonoid glycoside with different protective effects against lipid peroxidation and oxidative stress-mediated diseases [16]. Rutin has a protective effect against toxins on the ocular surface, according to Nebbioso et al. [17]. In addition, scientific evidence shows that rutin exhibits a cardioprotective effect against ISO-induced cardiotoxicity in rats [18,19]. Rutin has also been reported to be effective against diabetes [20], neurodegeneration [21], psoriasis [22], osteoarthritis [23], neuroinflammation [24], postoperative cognitive dysfunction [25], and so on. However, a detailed mechanism by which rutin achieves cardio-protection in ISO-induced toxicity has not yet been reported. Hence, the present study aims to examine the effect of rutin on the Nrf2/HO-1 signaling pathway in ISO-induced injury on rabbit corneal epithelial cells and in mice. The results demonstrated the preventive effectiveness of rutin toward ISO-induced corneal degeneration and heart damage through the management of oxidative stress.

## 2. Results

### 2.1. Effects of Rutin on Corneal Cells

The lysosomal accumulation of NR uptake in the rutin treatment was significantly similar to the pattern of the atorvastatin treatment (Figure 1b,d). The ISO-challenged cells showed decreased NR uptake. MTT assays revealed that rutin was the least toxic, making it the most promising preparation for potential clinical use. At the tested concentrations, the viability of the corneal epithelial cells was significant at ≤88%. The ISO treatment increased cell damage and death, which nonetheless recovered in both the rutin and atorvastatin treatment groups (Figure 1). However, an MTT assay showed that recovery was better with rutin when compared with atorvastatin, whereas an NR assay did not indicate any significant difference in ISO-induced cell toxicity between the rutin and atorvastatin groups. Treatment with rutin (20 µM) or atorvastatin (1 µM) alone did not show any toxicity toward the SIRC cell line, having 99% and 95% cell viability.

### 2.2. Effects of Rutin on Isoproterenol (ISO)-Challenged Corneal Cells

Through spectrophotometrically examining nitrate, the stable end byproduct of NO, we found that the ISO treatment increased the inflammatory reaction in corneal cells. The CO treatment with rutin and atorvastatin significantly reduced (*p* < 0.05) NO release. Both rutin and atorvastatin markedly decreased ISO-mediated NO generation in the SRIC cell lines (Figure 2a). In addition, treatment with rutin or atorvastatin significantly increased the expression of retinoic acid signaling components ADH7 and ALDH1A1 in SIRC rabbit corneal cell lines (Figure 2b).

### 2.3. Effects of Rutin on Cardiac Biomarkers in MI in Mice

Figure 3 shows the effects of rutin on cardiac marker enzyme levels in the control, ISO-induced MI, and rutin- or atorvastatin-pretreated mice. Creatine kinase-MB (CK-MB), cardiac troponin I (cTnI), cardiac troponin T (cTnT), and lactate dehydrogenase (LDH) were significantly elevated (*p* < 0.05) in mice administered ISO. Conversely, pretreatment with rutin or atorvastatin decreased CK-MB, cTnI, cTnT, and LDH levels.

### 2.4. Effect of Rutin on Oxidative Stress Markers in MI in Mice

Figure 4 shows the effects of rutin on lipid peroxidative marker levels and antioxidant status in the control, ISO-induced MI, and rutin- or atorvastatin-pretreated mice. Compared with the control mice, MDA levels were significantly elevated (*p* < 0.05) in mice induced with ISO. Pretreatment with rutin or atorvastatin significantly decreased (*p* < 0.05) ISO-induced MDA levels. Compared with the control mice, antioxidant enzyme activity levels (SOD, CAT, and GSH) were considerably decreased in the ISO-induced MI animals (*p* < 0.05). Compared with the ISO-induced mice, the SOD, CAT, and GSH activities were considerably elevated (*p* < 0.05) in MI mice that were given rutin or atorvastatin.

### 2.5. Effects of Rutin on Inflammatory Markers in ISO-Induced MI Mice

The inflammatory marker levels in the serum of the control, ISO-induced MI, and rutin- or atorvastatin-pretreated mice are shown in Table 1. Compared with the control mice, ISO-administered mice had significantly increased (*p* < 0.05) TNF-α, IL-6, and NF-κB levels. Conversely, pretreatment with rutin or atorvastatin inhibited ISO-induced increases in TNF-α, IL-6, and NF-κB.

### 2.6. Effects of Rutin on Apoptosis Marker Expression in the Hearts of ISO-Induced Mice

Figure 5 depicts the apoptosis marker expression in the hearts of the control, ISO-induced MI, and rutin- or atorvastatin-pretreated mice. Administering ISO to mice increased (*p* < 0.05) the expressions of myocardial pro-apoptotic signaling genes, including Bax and caspase 3, and decreased (*p* < 0.05) the gene expression of anti-apoptotic Bcl-2 (Figure 5a–c). Pretreatment with rutin or atorvastatin reversed the ISO-induced expression of both apoptotic and anti-apoptotic genes. Pretreatment with rutin or atorvastatin also significantly reduced (*p* < 0.05) caspase 3 activity (Figure 5d).

### 2.7. Effects of Rutin on Nrf2 Signaling in ISO-Induced Mice

Figure 6 depicts the protein expression of Keap1, Nrf2 (cytoplasmic and nuclear), Sirt1, and HO-1 in the hearts of the control, ISO-induced MI, and rutin- or atorvastatin-treated mice. The protein expression of cytoplasmic Nrf2 and Keap1 was elevated in the ISO-induced mice compared with the control mice, but there was no change in nuclear Nrf2 and Sirt1. Conversely, the ISO treatment decreased the expression of HO-1. Although the rutin and atorvastatin pretreatments increased nuclear HO-1, Keap1, and Sirt1 (*p* < 0.05) in ISO-induced mice, the rutin pretreatment alone increased nuclear Nrf2 expression (*p* < 0.05) in ISO-induced mice; the rutin pretreatment also decreased their cytoplasmic Nrf2 levels and increased their nuclear Nrf2 levels.

### 2.8. Effects of Rutin on Cardiac Histology in ISO-Induced Mice

Figure 7 shows histopathological alterations in mouse cardiac tissue exposed to rutin. The sections of the hearts of the control mice (Figure 7a) and mice treated with rutin or atorvastatin (Figure 7c,d) exhibited normal architecture and myofibrillar arrangements. When comparing the ISO group with the control group, we saw necrosis, widespread myofibrillar degenerative changes, a diffuse inflammatory process, and substantial myocardial injury (Figure 7b). However, microscopic examinations of the cardiac tissue revealed that the rutin pretreatment attenuated ISO-induced myocardial injuries (Figure 7c).

### 2.9. Molecular Docking Study

Rutin adhered to Keap1 because it was positioned at the hydrophobic pocket of the protein (Figure 8). The hydrophobic binding was stable because it was surrounded by the residues VAL-608, VAL-369, THR-560, ILE-559, VAL-606, VAL-514, VAL-467, GLY-367, and ALA-510. We also noticed that the hydrogen bond length of the rutin–Keap1 interaction ranged from 1.57 Å to 2.89 Å. The rutin–Keap interaction involved hydrophobic and non-conventional carbon–hydrogen bonds; the estimated binding energy of the rutin and Keap1 complex was −8.69 kcal/mol (Appendix A). In addition to nine conventional hydrogen bonds, two carbon–hydrogen bonds involving THR′560 and GLY′464′ with bond lengths of 2.54 Å and 3.19 Å, respectively, were also noticed in the rutin–Keap1 interaction. The results showed that rutin prevented Keap1 from interacting with Nrf2, preventing Nrf2 from breaking down and increasing its nuclear accumulation. The antioxidant capacity of the ISO-treated corneal cell line and heart tissue consequently increased as rutin promoted Nrf2/ARE signaling, possibly by opposing Keap1.

## 3. Discussion

Corneal degeneration can lead to functional impairment. This affects the ability to focus due to accumulated substances that cause inflammation and generate ROS, sometimes with severe pain [2]. Consequently, this can progress to vision loss and morbidity if not addressed in a timely manner [1]. Therefore, to prevent blindness and disability, treatment is imperative. However, treatment is usually performed via surgery and, in some cases, corneal transplantation [26].

In the present study, the use of ISO is based on the premise that large doses produce reactive oxygen species (ROS), which are implicated in oxidative stress-induced inflammation and subendocardial myocardial ischemia, hypoxia, and necrosis, serious stresses on the eye and myocardium [2,27]. Oxidative stress in corneal diseases occurs because ROS generated by external noxae are responsible for changes in the optical ability of the eyes, as well as reduced visual acuity and loss [28]. Accordingly, the integrity and function of the cardiac membranes are jeopardized by altered membrane permeability brought on by ISO-induced myocardial infarctions [29].

The present study showed that the rutin treatment mitigated the effects of ISO-induced corneal injury. However, rutin was better able to increase ADH7 expression in ISO-induced corneal damage than ALDH1A1 compared with atorvastatin. Therefore, the beneficial effects of statins, as documented in the literature, were confirmed in our study [30]. Western blotting analysis revealed strong ADH7 and ALDH1A1 protein expression after the rutin treatment, which was markedly absent with ISO exposure. This increased protein expression represents a direct and indirect retinoic acid signaling control, protecting the cornea from ROS and, hence, oxidative stress [31,32]. Therefore, a lack of retinoic acid or any condition that depletes the eye’s retinoic acid could disable focusing and vision because of injury [33]. Regarding effects on the heart, mice given ISO had higher circulatory cardiac biomarker levels, including CK-MB, cardiac troponin I, and cardiac troponin T, which may have been a sign of cell membrane necrosis or leakage and loss of functional integrity. In the rutin-pretreated mice, cardiac troponin I, cardiac troponin T, and CK-MB serum levels were significantly lower than in the ISO-induced mice. The isoenzyme activity of creatine kinase-MB can help to identify any kind of cardiac damage, and serves as an early diagnostic marker for myocardial infarction. Against isoproterenol-produced cardiotoxicity in mice, the rutin pretreatment reduced CK-MB activity and mediated the cardioprotective effect [19]. We also noticed increased blood cardiac troponin T and I levels in the mice with isoproterenol-induced cardiotoxicity, which has been previously described [34]. However, the rutin pretreatment reversed this ISO-mediated increase in both troponins. Additionally, pathological changes such as myocardial degeneration, severe cytoplasmic vacuolization, loss of myofibrillar alignment, and inflammatory cell infiltration improved. This suggests that rutin plays a cardioprotective role by reducing the release of myocardial enzymes and restoring the structure of cardiac tissues. This event is supported by previous reports on the cardioprotective effect of rutin in ISO-induced rats [8,18,29].

A growing body of research suggests that ISO causes oxidative stress by inducing reactive oxygen species (ROS). Previous research suggests that one potential molecular mechanism underlying ISO-induced myocardial infarction is the disruption of the delicate oxidant–antioxidant balance, which might result in myocardial injury via oxidative damage [35,36,37]. In the current study, mice injected with ISO had higher malondialdehyde (MDA) levels and lower levels of several cardiac antioxidants (SOD, CAT, and GPx) owing to increased free radical production and subsequent myocardiocyte lipid peroxidation. By contrast, mice fed with rutin could significantly enhance their antioxidant status (oppose excess free radicals) and reduce MDA formation, hence enhancing myocardiocyte integrity by inhibiting lipid peroxidation. The present findings agree with a previous study that found that flavonoid glycoside-like rutin possesses antioxidant activity with protective potential [38,39]. Elevations in pro-inflammatory indicators like NF-kB and TNF-α—along with the activation of oxidative stress in myocardial injuries—result in apoptosis and heart contractile dysfunction, occasionally leading to cardiac failure [40].

Apoptosis has been observed in numerous cardiovascular disorders, including myocardial infarction, hypertrophy, and heart failure, according to numerous lines of evidence. An anti-apoptotic protein called Bcl-2 binds to the mitochondrial membrane to prevent the release of cytochrome-C and cellular death [41]. However, Bax, a pro-apoptotic molecule, is dormant in healthy cells and promotes cell death. Bax goes through conformational activation in response to apoptotic stimuli like MI, which prevents Bcl-2 from taking an anti-apoptotic action [42]. In the current study, cleaved caspase 3 activity was higher in ISO-induced MI mice, but Bcl-2 mRNA expression levels were lower, indicating an apoptotic condition inside the myocardium. However, the rutin pretreatment reduced the observed alterations in Bax, caspase 3, and Bcl-2 levels, showing cardiac-cell-apoptosis-preventive efficacy.

The histopathological results were consistent with previous findings showing that ISO therapy disrupts cardiac muscle fibers, resulting in red blood cell extravasation and inflammatory cell buildup between the divided fibers. Increased cytotoxic free radical generation—which triggers lipid peroxidation and modifies the cardiac membrane—is thought to be responsible for degenerative alterations linked to ISO treatment [43,44]. Nevertheless, the deleterious effects of ISO therapy were reversed by the rutin pretreatment. Earlier studies have reported rutin’s ameliorating potential against cisplatin-induced renal damage, showing that administering rutin with cisplatin decreases abnormal tissue architectural features in mice [45]. This view was expressed by Arjumand et al., describing the restoration of renal histopathological changes in cisplatin–renal injuries in rats [46]. In another report, rutin inhibited tau-aggregation and tau-oligomer-induced cytotoxicity, protecting neuronal morphology by reducing pathological tau levels in mouse brains [47]. Du et al. [48] recently documented that rutin prevents neuronal loss by reducing SOD1-G93 in mice with amyotrophic lateral sclerosis. These findings appear to agree with the results of our study, in that rutin showed better histopathological features than the ISO treatment.

Molecular docking is a known useful computational approach and defined technique employed to predict the binding affinity of drugs to their target protein usually receptors. Therefore, its importance as a drug discovery tool is that it helps to develop new therapies for disease management. This technique was used to further confirm the extent of rutin and Nrf2 interaction.

Hence, the molecular docking results from the present study revealed the involvement of VAL-369, VAL-514, and VAL-608. The estimated binding energy of the rutin and Keap1 complex (−8.69 kcal/mol) was stronger than the Notoginsenoside and Keap1 complex (−7.75 kcal/mol) [48]. In addition, earlier reports showed that VAL-369, VAL-514, and VAL-608 are potentially involved in ligand interactions with Keap1 [49,50]. Previous research has shown that fumarate derivatives, which can bind to Keap1’s VAL-369 and VAL-608, can successfully displace the Nrf2 peptide from its binding site, and suggests that fumarate directly inhibits Keap1–Nrf2 binding [49]. These findings suggest that rutin inhibits the interaction between Keap1 and Nrf2, preventing Nrf2 from degrading and building up nuclear material. Consequently, the transcription of antioxidant enzymes downstream of Nrf2 is upregulated, lowering oxidative stress. Nrf2 is an essential participant in the oxidative stress pathway [51]; by boosting various antioxidant agents and enzymes, it can regulate the anti-inflammatory response and oxidative homeostasis [17,52].

The cornea is a constant target of inflammation owing to ROS generation; hence, to protect itself, it activates an antioxidant defense system controlled by Nrf2 [15]. Evidence shows that Nrf2 knockout mice exhibit a significant delay in wound healing [53]. Therefore, Nrf2 defense system activators may be useful agents in protecting against inflammatory conditions. This is supported by a recent study that showed the ability of rutin to enhance Nrf2 signaling, thereby reducing the inflammation induced by oxidative stress [54].

In this work, ISO-induced cardiac toxicity in mice showed a slight increase in Nrf2 activation because of ROS, which can activate Nrf2. This finding is consistent with earlier research showing that mitochondria-derived ROS activate Nrf2 [55]. ISO-induced animals also show a substantial increase in Keap1 expression. This suggests that Nrf2 is more linked to Keap1, reducing Nrf2 translocation in the nucleus to activate antioxidant gene expression. However, the rutin treatment drastically reduced Keap1 protein expression, as well as significant nuclear accumulations of Nrf2 in mouse myocardia. Under normal settings, the nuclear factor Nrf2 binds to Keap1 in the cytoplasm. When exposed to oxidative stress or other potentially harmful stimuli, Nrf2 dissociates from Keap1 and translocates into the nucleus, where it binds to antioxidant response element (ARE) sequences, activating the transcription of phase II enzymes/antioxidant genes such as HO-1. In this study, rutin also increased HO-1 in ISO-induced cardiac tissue. Further docking studies show that rutin binds to Keap1, suggesting that it may boost Nrf2 concentrations in the nucleus, encouraging the transcription of important antioxidant and phase II enzymes. Our results thus demonstrate that administering rutin increases Nrf2-mediated antioxidant responses and that phase II enzymes may help to protect cells from oxidative stress and improve their ability to defend themselves against free radical damage.

## 4. Materials and Methods

### 4.1. Drugs, Chemicals, and Reagents

Rutin and isoproterenol were purchased from Sigma Aldrich (Steinheim, Germany). SOD, Catalase, MDA, and GSH assay kits were purchased from Sigma-Aldrich (Steinheim, Germany). LDH and creatine kinase (CK-MB) isoenzyme were acquired from Accurex Biomedical Pvt. Ltd. (Mumbai, India). A troponin-I and -T detection kit was purchased from Abcam (Santa Cruz, CA, USA), who also supplied the primary and secondary antibodies used in this investigation. The additional chemicals and reagents with the best analytical grade were all obtained from Himeda (Mumbai, India) and Merck (Bengaluru, India).

### 4.2. Rabbit Corneal Cell Culture Preparation

The experiments were conducted using the normal rabbit corneal epithelial cell line SIRC1 (NCCS, Pune, India). The cells were grown as monolayers in 25 cm^2^ culture flasks (Tarsons, Kolkata, India) in DMEM medium (Himedia, Mumbai, India) supplemented with antibiotics (100 U/mL penicillin, 100 µg/mL streptomycin; Himedia, Mumbai, India) and 10% FBS (Himedia, Mumbai, India). The cells were incubated at 37 °C in a humidified atmosphere with 5% CO_2_.

### 4.3. MTT Assay

The total number of cells for the experiments was calculated using a Thomas hemocytometer (Deluxe Scientific Surgico PVT LTD, New Delhi, India). In 96-well flat-bottomed cell culture plates, 100 µL of cell suspension (1 × 10^4^ cells/mL) was added. After 24 h, the medium was discarded and pretreated with rutin (20 µM) or atorvastatin (1 µM) for 6 h in serum-free DMEM. Then, the appropriate groups were treated with ISO (50 µM) for 24 h. As controls, untreated cells were cultured in 100 µL of medium with a total cell number equivalent to that of the sample wells and incubated for 24 h. A blank control was simply culture medium. After a 24 h incubation period, cytotoxicity was determined using spectrophotometric methods; namely, the MTT [56] and NR assays [57].

### 4.4. Neutral Red (NR) Uptake Assay

The cell line was plated individually using 96-well plates at a concentration of 1 × 10^4^ cells/well in DMEM media containing 10% fetal bovine serum (Himedia, Mumbai, India) and 1× antibiotic antimycotic solution in a CO_2_-incubator in a 37 °C environment with 5% CO_2_. After being cleaned with 200 μL of 1× PBS, the cells were treated in the MTT assay and allowed to incubate for 24 h. When the treatment period ended, the media were aspirated from the cells. Neutral red stain (40 μg/mL) in 1× PBS was added to the wells and incubated for 4 h at 37 °C. Following the incubation period, the NR-containing media were removed from the cells and twice cleaned with 150 μL of PBS. To eliminate precipitated dye crystals, we added 150 μL of destain solution to each well and vortexed the plate for ten minutes. Using cell-free blanks as a reference, color intensity development was measured at 540 nm on a microtiter plate reader [57].

### 4.5. Nitric Oxide (NO) Measurement

In culture supernatants, a spectrophotometric technique based on the Griess reaction was used to measure nitrate, a stable byproduct of NO. The nitrite level indicates NO production [58]. In the present investigation, isoproterenol (50 µM), rutin (20 µM), and atorvastatin (1 µM) were incubated for 24 h on corneal epithelial cells. After 24 h of incubation, the treated cells’ culture supernatants were examined for NO. Utilizing a microplate reader (Molecular Devices Corp., Emax, Menlo Park, CA, USA), the optical density of the samples and 0.5–25 µM of sodium nitrite (NaNO_2_) standards were determined at 550 nm. The final values are presented in µM.

### 4.6. Animals

This study included thirty-two male Swiss albino mice, weighing 20 ± 2 g and being four weeks old, acquired from the animal facility of King Faisal University (KFU), located in Al-Ahsa, Saudi Arabia. Guidelines for the use and care of experimental mice were adhered to in all animal research according to the Ethical Council of Deanship of Scientific Research (KFU-REC-2021-OCT-EA1831). Each mouse was housed in a controlled environment with a 12 h light–dark cycle, 45–60% relative humidity, and a constant temperature of 23 ± 2 °C. For a week, all mice received adaptive feeding, meaning they were free to eat and drink. King Faisal University’s Institutional Animal Treatment and Use Committee (IACUC) approved all animal experiments and treatments.

### 4.7. Experimental Design

A total of 32 male Swiss albino mice were split up into 4 groups, with 8 in each group. Group I was the control group, given a standard pellet diet; Group II was the disease group, injected with ISO; Group III was treated with rutin (20 mg/kg body weight/day) using an oral gavage; and Group IV was the drug control group, pretreated with atorvastatin (10 mg/kg body weight/day) using an oral gavage. As previously mentioned [59], myocardial infarction in Groups II to IV was caused by receiving subcutaneous injections of ISO at 85 mg/kg of body weight (ISO dissolved in saline) at 24 h intervals during the last two days (the 13th and 14th) of the experiment, while control Group I received sterile saline.

At the end of the experiment period (15th day), blood was collected through cardiac puncture under a combination anesthetic with ketamine (75 mg/kg) and xylazine (10 mg/kg), and the mice were sacrificed via cervical dislocation. Blood was separated via centrifuge at 1109 g for 15 min, and the serum was used to investigate cardiac markers. The heart (50 mg) was removed and homogenized with 1 mL of ice-cold Tris-HCl buffer (pH 7.4) and then centrifuged, and the clear supernatant was separated. The separated supernatant was used to determine the heart’s antioxidant and inflammatory marker levels.

### 4.8. Measurement of cTnI, CK-MB, and LDH

CK-MB and LDH activities were measured using AutoPure CK-MB 25 and a liquid LDH 10 diagnostic kit, respectively (Accurex Biomedical Pvt. Ltd., Mumbai, India), while serum cTnI and cTnT were measured with a mouse troponin I ELISA kit and a mouse troponin I ELISA kit, respectively (Abcam, Waltham, MA, USA). Each assay was conducted according to the manufacturer’s instructions.

### 4.9. Determination of the Antioxidant Level

The antioxidant system assay was performed by estimating the amounts of reduced glutathione (GSH), catalase (CAT), malondialdehyde (MDA), and superoxide dismutase (SOD) in serum using ready-to-use kits (Sigma Aldrich, Steinheim, Germany).

### 4.10. Determination of Pro-Inflammatory Markers

Using specialized ELISA kits provided by Shanghai Crystal Day Biotech Co., Ltd., Shanghai, China, Bioassay Technology Laboratory, TNF-α, IL-6, and NF-κB levels were measured in the heart tissue. Each assay was carried out according to the manufacturer’s instructions.

### 4.11. Caspase 3 Assay

Caspase 3 activity was quantified using a Caspase 3 Assay Kit (Colorimetric) (Make: Abbkine, Wuhan, Hubei, China; Cat No: KTA3022) per the manufacturer’s instructions. In brief, 20 mg of tissue was washed with PBS and homogenized using 0.1 mL of Working Cell Lysis Buffer. Then, the lysate was incubated on ice for 15–20 min and centrifuged at 17,700× *g* for 15 min at 4 °C, and the supernatant was transferred to a new tube. Protein content in the lysate was quantified, and the remaining protein suspension was used for a caspase 3 calorimetric assay. In total, 50 µL of 1× Reaction Buffer (containing DTT) was added to each 50 µL sample (samples, background control, and standard samples). Then, 5 μL of the 4 mM Ac-DEVD-pNA substrate (200 μM final concentration) was added to each sample, mixed well, and incubated at 37 °C for 1–2 h. The absorbance of each well was recorded at 405 nm on a microtiter plate reader (Alere, India; Model: AM 2100). The caspase 3 activity was calculated in µmol pNA released per minute per ml of cell lysate using the formula µmol pNA/(*t* × *v*), where, *v* = volume of sample in mL and *t* = reaction time in minutes. Finally, the results were expressed as pMol pNA/min/mg of protein.

### 4.12. Quantitative Real-Time PCR

TRIzolTM was used to isolate total RNA from the heart according to the manufacturer’s instructions (Takara, New Delhi, India). Using the RNA PrimeScriptTM 1st strand cDNA Synthesis Kit (Takara, New Delhi, India) and adhering to the manufacturer’s instructions, cDNA was reverse-transcribed from 1 μg. Quantitative RT PCR was performed using a Rotor-Gene Q 2PLEX HRM Real-Time PCR system (Qiagen, Hilden, Germany). Relative levels of mRNA transcripts for bcl2, Bax, and caspase 3 were quantified via qRT-PCR with TB Green Premix Ex Taq II (Takara, New Delhi, India). A list of primers used for qRT-PCR is included in Table 2. Using the ΔΔCt approach, transcripts of glyceraldehyde 3-phosphate dehydrogenase (GAPDH) were utilized to normalize the target genes’ expression levels [60].

### 4.13. Western Blot Analysis

An ice-cold RIPA buffer was used to extract the total protein from the cardiac tissue specimens and rabbit corneal cell line (SIRC1). The homogenate was then centrifuged at 17,700× *g* for 30 min at 4 °C to dispose of any remaining debris. Using the PierceTM BCA protein assay kit (Thermo Fisher Scientific, Rockford, IL, USA), the protein content of the samples was estimated. SDS polyacrylamide (12%) was used to load and separate the protein samples. To mitigate non-specific binding sites, the membranes were treated for one hour with blocking buffer containing 5% bovine serum albumin. After the proteins were placed on a PVDF membrane, they were incubated overnight at 4 °C with gentle shaking. The primary antibodies used were Keap1 (1:1000) from Sigma, St. Louis, MO, USA; Nrf2 (1:1000), Sirt1 (1:1000), and HO-1 (1:500) from Santa Cruz Biotechnology, Dallas, TX, USA; and ADH7 (1:500), ALDH1A1 (1:500), and β-actin/tubulin/laminin (1:2000) from MERCK Millipore, USA. Following this, the membranes were incubated for 1 h at room temperature with their corresponding secondary antibodies (anti-mouse or anti-rabbit IgG conjugated to horseradish peroxidase). The protein bands were then visible under an enhanced chemiluminescence pico kit (Thermo Fisher Scientific, Rockford, IL, USA). Image J (https://imagej.net/ij/, accessed on 10 September 2024, NIH, Bethesda, MD, USA) was used to quantify the intensity of the generated bands.

### 4.14. Histopathology of Myocardium

After the hearts were removed, they were promptly cleaned with 0.9% saline and preserved in 10% formalin. Paraffin was used to embed and prepare fixed cardiac tissues. Following paraffin block sectioning, hematoxylin-eosin staining was applied. Under an optical microscope, the stained sections were inspected [61]. The histological observations were evaluated based on the presence and severity of the following criteria: muscle necrosis, leukocytic infiltration, and persistent inflammation. Semi-quantitatively, each ingredient was marked using a scale, with 0 being normal and 1–4 representing a score ranging from modest infiltration and inflammation to substantial changes.

### 4.15. Molecular Docking Analysis

The Autodock v4.2 and Auto dock tools (ADT) v1.5.4 were used to perform the docking analysis [62]. Rutin’s 3D chemical structure (CID_5280805) was retrieved from the PubChem database. Keap (PDB ID: 2w96_B) for mice was obtained from http://www.pdb.org (accessed on 27 August 2023). Target protein complexes were docked, with the ligand having a flexible body and the molecule having a rigid body. The whole receptor amino acid employed for blind docking was searched. The Lamarckian Genetic Algorithm was used to conduct this search, with 150-person populations with a 0.02 mutation rate evolved over the course of five generations. The various complexes were sorted according to the anticipated binding energy to assess the outcomes. After that, a cluster analysis was carried out using root mean square deviation values in relation to the initial geometry. The most reliable solution was determined to be the one with the lowest energy conformation of the most populous cluster. A trial version of the Discovery Studio 2021 Client was used to observe and analyze the docked ligand–receptor interactions.

### 4.16. Statistical Analysis

The means ± standard error of the means (SEMs) represent all generated values. The GraphPad Prism 10.2 statistical software was employed: one-way ANOVA was used for statistical comparisons between groups and among groups, and Tukey’s post hoc test was used afterward. A *p*-value of less than 0.05 was deemed statistically significant.

## 5. Conclusions

In summary, our study showed that rutin significantly protected rabbit corneal epithelial cells from ISO-induced oxidative stress. In addition, it upregulated the expression levels of ADH7 and ALDH1A1, which are responsible for increased levels of retinoic acid, an important agent in corneal homeostasis. In effect, this could function as a protection against cornea injury due to inflammation. We also observed a cardioprotective property of rutin against ISO-induced myocardial infarction in a mouse model. Nrf2 signaling was activated by rutin and mediated its protective effect against ISO-induced myocardial infarction. The increase in antioxidant activity might be attributable to rutin-mediated Nrf2 antioxidant defense systems.

## Figures and Tables

**Figure 1 pharmaceuticals-17-01523-f001:**
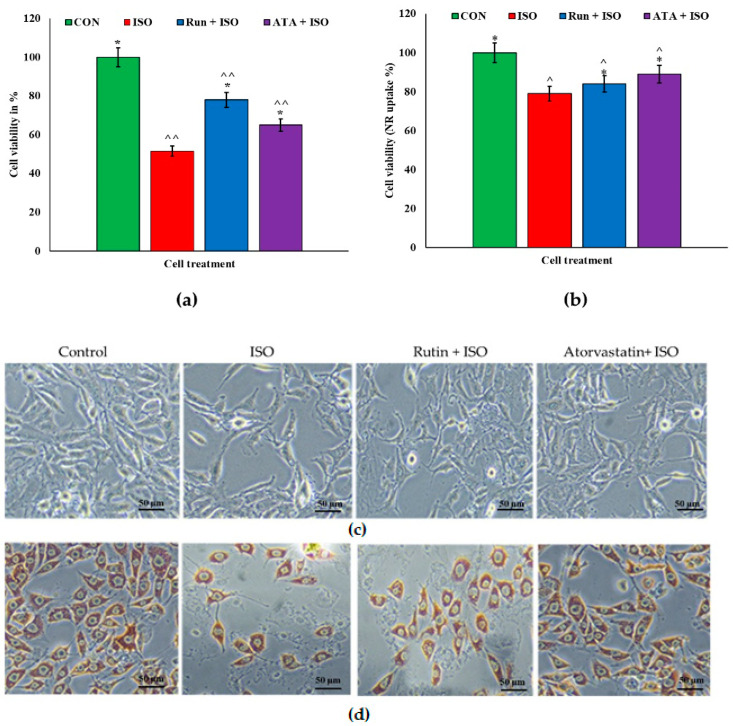
MTT and neutral red assay showing the effects of rutin and atorvastatin on rabbit corneal epithelial cell viability. Rabbit corneal epithelial cells (SIRC1) were subjected to pretreatment with either rutin (20 µM) or atorvastatin (1 µM) for 6 h and subsequently treated with ISO (isoproterenol) (50 µM) for an additional 24 h. The results were compared with cultures that were not treated and ISO-challenged cells. The MTT assay (**a**) and the neutral red assay (NR) (**b**) present an average of three separate experiments; the results are noted as a percentage of the controls, which is arbitrarily set at 100%. (**c**) Phase contrast images of SIRC cell lines acquired before adding the MTT reagent. The bars represent the mean ± SEM for n = 3. (**d**) Phase contrast images of neutral red-stained SIRC cell lines. Phase contrast images were acquired using a 20× objective lens and an inverted phase contrast microscope (Optika, IM-3FL4, Ponteranica (BG), Italy). The results were compared with wells that were not treated and ISO-challenged cells. Values are mean ± SEM for each group. * *p* < 0.05 versus ISO group; ^ *p* < 0.05, and ^^ *p* < 0.01 versus the control group, n = 3.

**Figure 2 pharmaceuticals-17-01523-f002:**
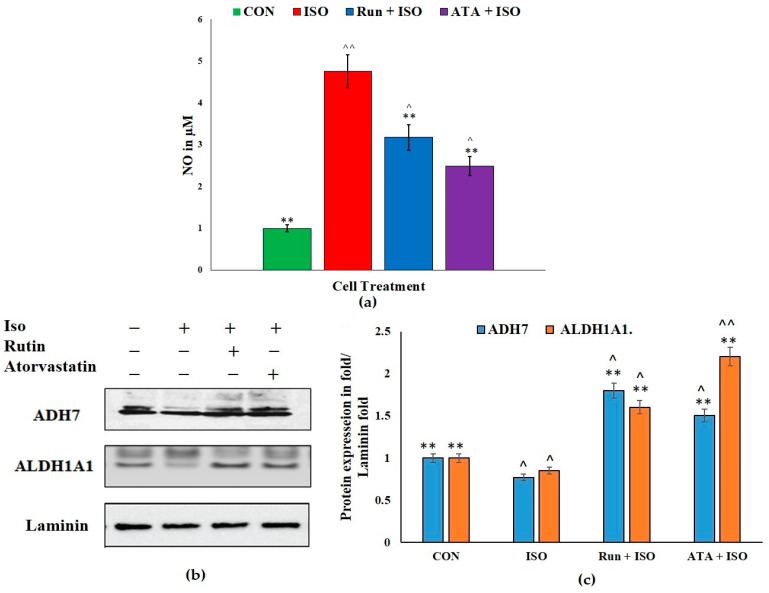
Quantification of ISO-induced NO generation in SIRC cell lines and ADH7 and ALDH1A1 expressions in rutin and atorvastatin treatments of ISO-induced corneal damage in SIRC1 cell lines. Rabbit corneal epithelial cells (SIRC1) were subjected to pretreatment with either rutin (20 µM) or atorvastatin (1 µM) for 6 h and subsequently treated with ISO (isoproterenol) (50 µM) for an additional 24 h. The results were compared with cultures that were not treated and ISO-challenged cells. (**a**) Nitric oxide (NO) secretion in cultured normal rabbit corneal epithelial cells. (**b**) Protein expressions of ADH7 and ALDH1A1 in the SIRC1 cell lines assessed using Western blotting analysis; (**c**) quantification of ADH7 and ALDH1A1 expression based on intensity of the blot. Rutin enhances the ADH7 and ALDH1A1 retinoic acid signaling components in the SIRC cell lines. (**b**,**c**), these depict the ratio of the relative amount of marker expression to aggregate in fold. Values are mean ± SEM for each group ** *p* < 0.01 versus ISO group; ^ *p* < 0.05, and ^^ *p* < 0.01 versus the control group, n = 3.

**Figure 3 pharmaceuticals-17-01523-f003:**
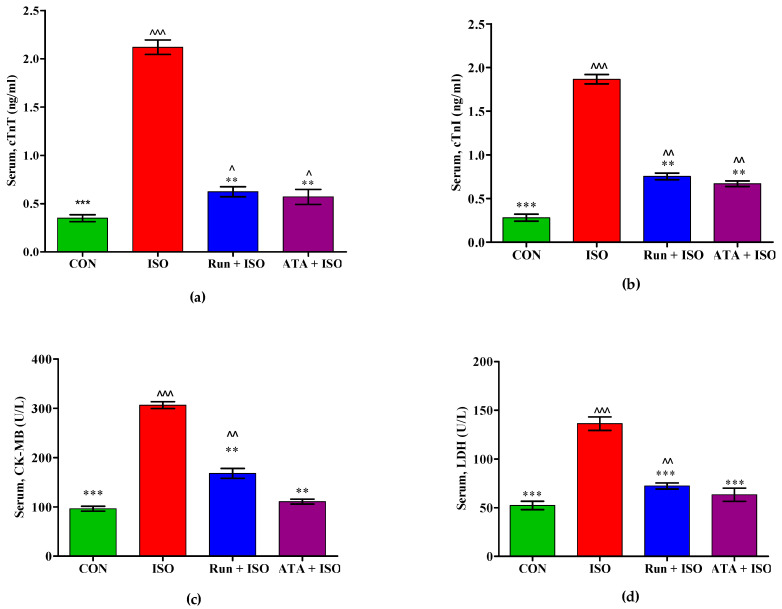
Effects of rutin and atorvastatin treatments on serum cardiac biomarkers in mice with ISO-induced cardiotoxicity. Mice were subjected to pretreatment with either rutin (20 mg/kg body weight/day) or atorvastatin (10 mg/kg body weight/day) for 14 days and subsequently treated with ISO (isoproterenol) (85 mg/kg of body weight) for the last 2 days. (**a**) Effect of rutin on serum cTnT; (**b**) effect of rutin on serum cTnI; (**c**) effect of rutin on serum CK-MB; (**d**) effect of rutin on serum LDH contents in serum. Values are mean ± SEM for each group. ** *p* < 0.05 and *** *p* < 0.01 versus ISO group; ^ *p* < 0.05, ^^ *p* < 0.01, and ^^^ *p* < 0.001 versus the control group, n = 8. cTnI = (cardiac troponin I), cTnT = (cardiac troponin T), LDH = (lactate dehydrogenase), and CK-MB = creatine kinase-MB.

**Figure 4 pharmaceuticals-17-01523-f004:**
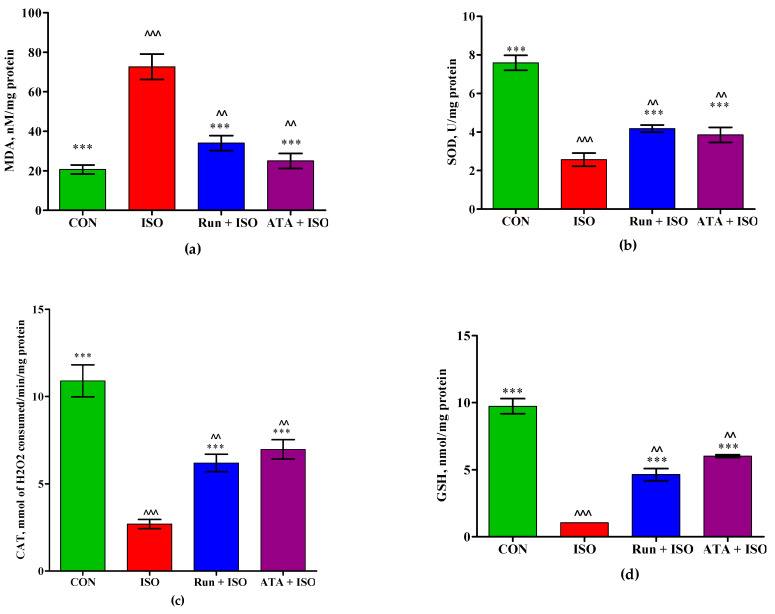
Effects of rutin and atorvastatin treatments on antioxidant system in mice with ISO-induced cardiotoxicity. Mice were subjected to pretreatment with either rutin (20 mg/kg body weight/day) or atorvastatin (10 mg/kg body weight/day) for 14 days and subsequently treated with ISO (isoproterenol) (85 mg/kg of body weight) for the last 2 days. (**a**) Effect of rutin on heart tissue MDA content; (**b**) effect of rutin on heart tissue SOD activity; (**c**) effect of rutin on heart tissue CAT activity; (**d**) effect of rutin on heart tissue GSH contents. Values are mean ± SEM for each group. *** *p* < 0.01 versus ISO group; ^^ *p* < 0.01, and ^^^ *p* < 0.001 versus the control group, n = 5.

**Figure 5 pharmaceuticals-17-01523-f005:**
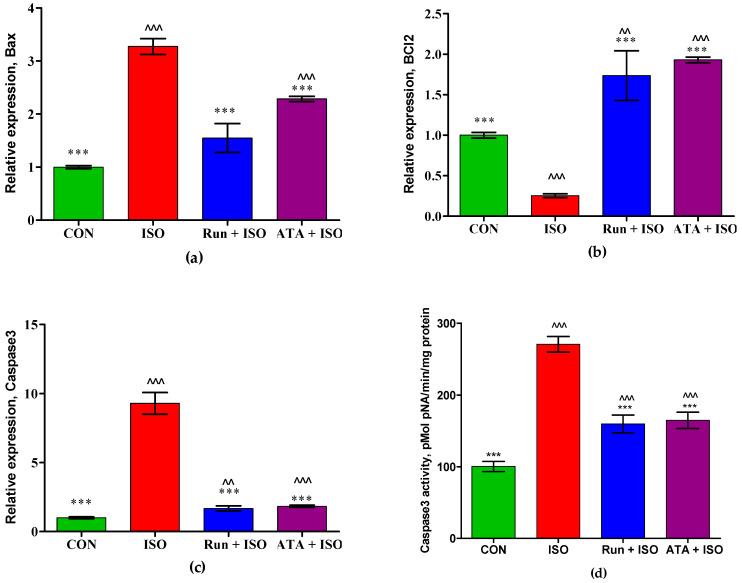
Effects of rutin and atorvastatin on relative mRNA expression of genes involved in the apoptotic process and caspase 3 activity in mice with ISO-induced cardiotoxicity. Mice were subjected to pretreatment with either rutin (20 mg/kg body weight/day) or atorvastatin (10 mg/kg body weight/day) for 14 days and subsequently treated with ISO (85 mg/kg of body weight) for the last *2* days. (**a**) Effect of rutin on expression of Bax mRNA; (**b**) effect of rutin on expression of bcl2 mRNA; (**c**) effect of rutin on expression of caspase 3 mRNA; (**d**) effect of rutin on caspase 3 activity. Values are mean ± SEM for each group. *** *p* < 0.01 versus ISO group; ^^ *p* < 0.01, and ^^^ *p* < 0.001 versus the control group, n = 5.

**Figure 6 pharmaceuticals-17-01523-f006:**
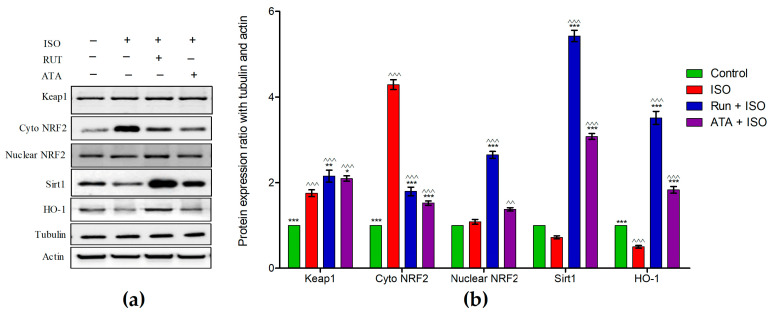
Effects of rutin and atorvastatin on relative protein expression in mice with ISO-induced cardiotoxicity. Mice were subjected to pretreatment with either rutin (20 mg/kg body weight/day) or atorvastatin (10 mg/kg body weight/day) for 14 days and subsequently treated with ISO (85 mg/kg of body weight) for the last 2 days. (**a**) The protein expressions of Keap1, cytoplasmic Nrf2, nuclear Nrf2, Sirt1, and HO-1 in heart tissues were assessed using Western blotting analysis. (**b**) Quantification of Keap1, cytoplasmic Nrf2, nuclear Nrf2, Sirt1, and HO-1 expression based on the intensity of the blot. Values are mean ± SEM for each group. * *p* < 0.05, ** *p* < 0.01, and *** *p* < 0.001 versus ISO group; ^^ *p* < 0.01, and ^^^ *p* < 0.001 versus the control group, n = 5.

**Figure 7 pharmaceuticals-17-01523-f007:**
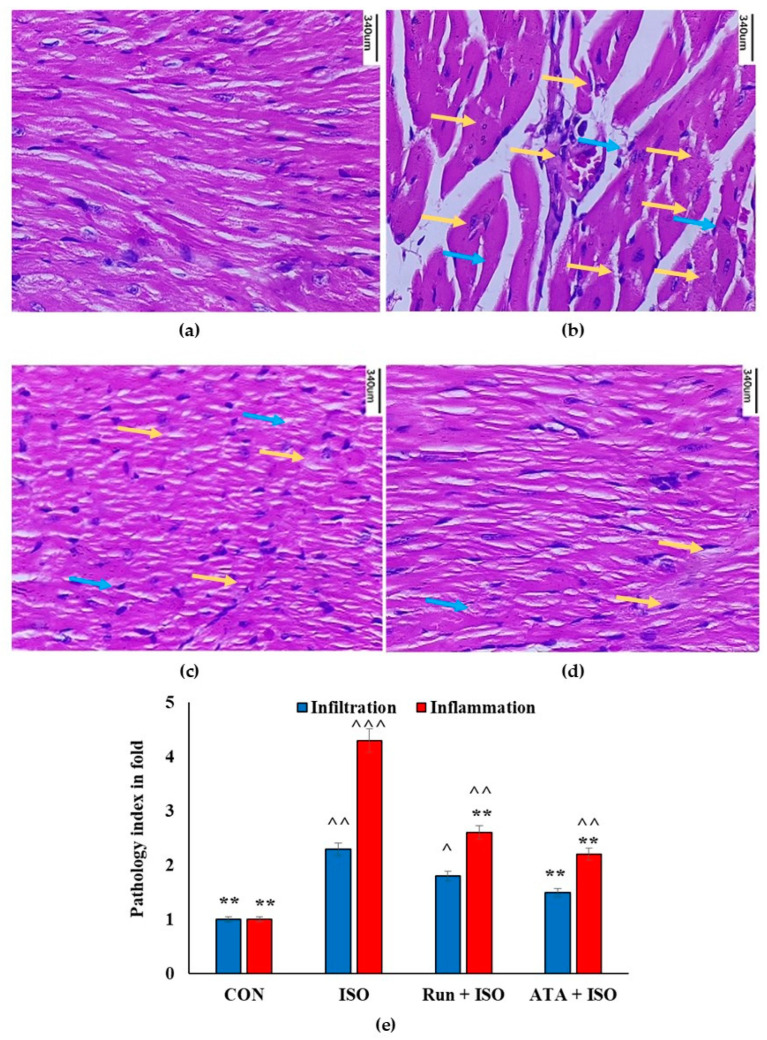
Effects of rutin and atorvastatin on cardiac histology in mice with ISO-induced cardiotoxicity. Mice were subjected to pretreatment with either rutin (20 mg/kg BW/day) or atorvastatin (10 mg/kg body weight/day) for 14 days and subsequently treated with ISO (85 mg/kg of body weight) for the last 2 days. (**a**) Photomicrographs of heart sections of control displaying normal cardiac cell architecture. (**b**) Photomicrographs of heart sections of ISO-treated mice showing distorted cardiac cells with many infiltrations. (**c**) Photomicrographs of heart sections of ISO + rutin-treated mice showing inflammation recovery. (**d**) Photomicrographs of heart sections of ISO + atorvastatin-treated mice showing better recovery from inflammation; (**e**) Pathology indexing of experimental group based on muscle necrosis, leukocytic infiltration, and inflammation. Slides were stained using hematoxylin and eosin (H&E) and imaged at ×200 using a triocular compound microscope (Labomed, Gurgaon, India, Lx 400) (scale bar = 340 µm). Blue arrow mark indicates infiltration and yellow arrow mark indicates inflammation. ** *p* < 0.05 versus ISO group; ^ *p* < 0.05, ^^ *p* < 0.01 and ^^^ *p* < 0.001 versus the control group, n = 3.

**Figure 8 pharmaceuticals-17-01523-f008:**
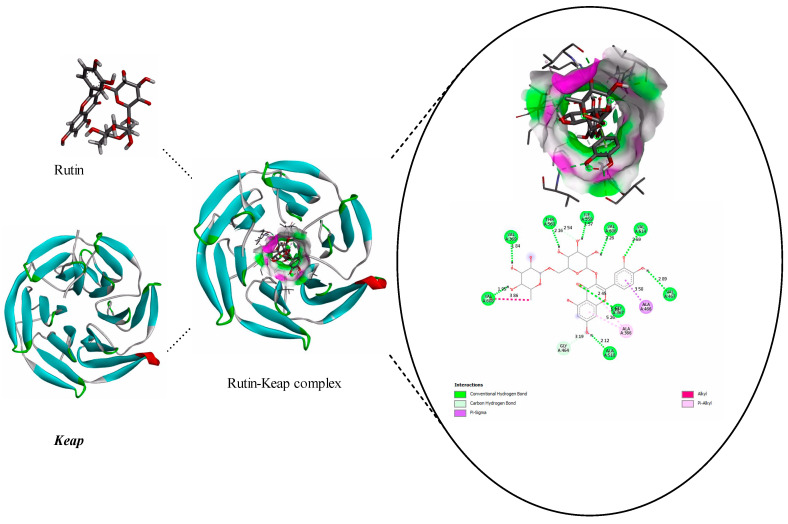
The bond between rutin and the Keap protein is stabilized by a conventional hydrogen bond and hydrophobic interactions. Green indicates the involvement of the hydrogen bond and pink indicates the involvement of the hydrophobic bond in the rutin–Keap interaction.

**Table 1 pharmaceuticals-17-01523-t001:** Effects of rutin and atorvastatin treatments on circulatory pro-inflammatory markers in mice with ISO-induced cardiotoxicity.

Groups	Control	ISO	Rutin + ISO	ATR + ISO
NF-κB (pg/mL)	24.72 ± 1.73 ***	75.28 ± 6.45 ^^	39.24 ± 2.46 ^^ **	37.48 ± 3.78 ^^ **
IL-6 (pg/mL)	68.42 ± 5.15 **	127.55 ± 10.47 ^^	87.48 ± 6.37 ^^ ***	79.32 ± 7.14 ^ **
TNF-α (pg/mL)	41.64 ± 4.04 **	75.49 ± 6.25 ^^	59.17 ± 4.58 ^ **	55.13 ± 4.47 ^ **

Values represent mean ± SEM for each group. ** *p* < 0.05 and *** *p* < 0.01 versus ISO group; ^ *p* < 0.05, and ^^ *p* < 0.01 versus the control group, n = 8. ISO—isoproterenol.

**Table 2 pharmaceuticals-17-01523-t002:** List of primers used for qRT-PCR.

Sl No	Gene	Forward Primer (5′–3′)	Reverse Primer (5′–3′)	Product Size (bp)
1	*bcl-2*	GTGGATGACTGAGTACCT	CCAGGAGAAATCAAACAGAG	118
2	*Bax*	CTACAGGGTTTCATCCAG	CCAGTTCATCTCCAATTCG	133
3	*Caspase 3*	CCAACCTCAGAGAGACATTC	TTTCGGCTTTCCAGTCAGAC	254
4	*GAPDH*	GAGAAACCTGCCAAGTATG	GGAGTTGCTGTTGAAGTC	123

## Data Availability

The original contributions presented in the study are included in the article/supplementary material, further inquiries can be directed to the corresponding author/s.

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
