# Peer review of "Rutin-Activated Nuclear Factor Erythroid 2-Related Factor 2 (Nrf2) Attenuates Corneal and Heart Damage in Mice"

_pharmaceuticals, 2024, doi:10.3390/ph17111523_

Round 1
Reviewer 1 Report (New Reviewer)
Comments and Suggestions for Authors
The manuscript "Rutin activated Nuclear factor erythroid 2-related factor 2 (Nrf2) to attenuate cornea and heart damage in rats" is devoted to the study biological activity of rutin in relation to oxidative damage caused by isoproterenol, inflammation, on epithelial cells of the rabbit cornea and damage to the heart. MTT, neutral red uptake assays, nitric oxide production measurement, PCR and molecular docking methods were applied. The data obtained in the work have been well and thoroughly discussed.
The study may be useful for researchers working in the field of pharmaceutical science and biochemistry.
The manuscript is well structured, and the data is presented quite clearly. Thus, I think that the manuscript can be published in the Pharmaceutical journal in the present form.
Author Response
Please see the attachment

Reviewer 2 Report (New Reviewer)
Comments and Suggestions for Authors
I read the article of the respected Emeka et al. with great interest. I find the choice of pathologies interesting. Oxidative damage to the cornea and myocardium are the leading steps in the pathogenesis of myocardial infarction and cornea degeneration. I also find the choice of antioxidant valuable. Rutin is a very well-known flavonoid, which is used in the treatment of venous insufficiency. Therefore, it can be easily prescribed by practicing physicians. We know that oxidative damage is attenuated by antioxidants, but it is very important to know the mechanism of action. This mechanism obviously includes the activation of various signaling pathways. From this point of view, this study is a good contribution to oxidative biochemistry. I also liked the choice of oxidative stress inducer. Isoprenaline is an adrenaline derivative, which is similar to it in biological properties. I believe that the positive effect of rutin is broader, perhaps it also exhibits an anti-stress effect.
The study design includes in vitro and in vivo experiments. The authors conducted a comprehensive study, including histology, biomarkers, gene and protein expression, and molecular docking calculations. The study is carried out at a high scientific level. The conclusions are supported by the results. The choice of methods is adequate. I had only a few minor comments while reading.
1. Materials and methods, subsections 4.8 and 4.9. Please specify the names of the analytical kits and the manufacturers.
2. The lower boundaries of the error bars are almost indistinguishable in Fig. 1a, 1b, and 2a.
3. In Fig. 3, on the Y axis, the Latin word ‘sereum’ should be replaced with the English word ‘serum’.
4. Line 322, the sentence seems unfinished to me. Myocardial damage?
Author Response
Please see the attachment

Reviewer 3 Report (New Reviewer)
Comments and Suggestions for Authors
The current style of life is full of stress, pollution, highly converted/processed food lack of physical activity etc. All of the above leads to the dysfunction of the cardiovascular system which results in retinopathy, and whole CNS injury. One of the factors which lead to cell death and ageing is the ROS and RNS which exist in the normal cell in the balance with the antioxidation system enzymatic and non-enzymatic. The low-weight molecules such as routine play the support role for the endocellular system and protect the cell's external layer against harmful factors delivered with physiological fluids. Authors in their article entitled: Rutin activated Nuclear factor erythroid 2-related factor 2 (Nrf2) to attenuate cornea and heart damage in rats have taken into consideration the benefits derived directly from the rutin supplementation. Their studies have been performed in three directions: physicochemical, animal model and biochemical. The planned and described experimental methods were almost correctly described. I recommend putting more details in the case of molecular docking. Also, I recommend moving Table 2 to the supplementary materials and removing the green colour from the text. The article is well written and readable with correct reference selection and citation. Due to the high-quality results presentation and their significant meaning for human health due to the protection role of rutin on the cardiovascular system and against retinopathy, I can recommend the article for publication.
Author Response
Please see the attachment

Reviewer 4 Report (New Reviewer)
Comments and Suggestions for Authors
Rutin is a common dietary flavonoid and has been reported to exhibit significant pharmacological effects, including anti-oxidation and anti-inflammation, etc. This study examined the preventive effects of rutin on isoproterenol (ISO)-induced oxidative damage, inflammation, in rabbit corneal epithelial cell line SIRC1 in vitro and mice heart injury in vivo. Results revealed the preventive effectiveness of rutin towards ISO-induced corneal injury and heart damage by managing oxidative stress possibly via Kelch-like ECH associated protein 1 (Keap1)/nuclear factor erythroid 2-related factor 2 (Nrf2) pathway. There are many concerns as listed in the following, including errors, typos, and inconsistent writing type.
*L25: signalling components component
**L27, L29: NRF2 and L28: KEAP; L31: KEAP/NRF2 -> inconsistent writing with Nrf2 and Keap in the Text.
*L33: reactive oxygen specie-> reactive oxygen species
*L44-45: One of the many types of CVDs, myocardial infarction (MI), major cause of global disability and death, especially in poorer nations[6]. > no verb
L60: Ma, [14]
L63: reactive oxygen species (ROS) -> ROS
*L65-66: Rutin is a flavonoid glycoside with different protective effects against lipid peroxidation and oxidative-stress-mediated diseases has been described [16].
*L69, L73: isoproterenol-induced-> ISO-induced
L71: Osteoarthritis
*L72: ostoperative cognitive dysfunction->postoperative cognitive dysfunction -> postoperative
L77: corneal injury, and heart damage
**L80-81: The lysosomal accumulation was explored by NR uptake and rutin showed recovered lysosomal debris and granules in corneal cell and significantly similar to atorvastatin treatment. ->The data were shown in which figure?
**L84: MTT assays revealed that the Rutin was the least toxic, making it the most promising preparation for potential clinical use.?? [no data for Rutin or atorvastatin only were presented in Fig. 1A and 1B]
**L88: whereas NR assay did not indicate any significant difference in ISO-induced cell toxicity.?? -> but * was labelled on Run+ISO group and ATA+ISO group in Fig. 1b
*L91: Fig.1d: It is better to change the label ISO+Rutin to Rutin+ISO and ISO+ atorvastatin to atorvastatin+ISO
L98: The MTT assay (a) and the neutral red assay (NR). (b)
**L100: The bars represent the mean plus standard deviation for n = 3. Vs. L104: Values are mean ± SEM for each group.??
***L104-105: ** P < 0.05, *** P < 0.01 versus control group; ˄ P < 0.05, ˄˄ P < 0.01, ˄˄˄ P < 0.001 versus the ISO group, n = 3 ??-> * P < 0.05 versus ISO group; ˄ P < 0.05, ˄˄ P < 0.01 versus the control group, n = 3
**L107: Inflammatory cell infiltration was estimated by NO quantification. -> Rabbit SIRC1 cell line contains Inflammatory cells??
**L111: Both rutin and atorvastatin showed a stronger?? NO-reducing effect (Figure 2A).
L113: HCE rabbit corneal cell lines??
L117: Showing quantification of NO by inflammatory cells??
L126: (B and C) -> (b and c)
***L127: ** P < 0.05, *** P < 0.01 versus control group; ˄ P < 0.05, ˄˄ P < 0.01, ˄˄˄ P < 0.001 versus the ISO group, n = 3. ->** P < 0.01 versus ISO group; ˄ P < 0.05, ˄˄ P < 0.01 versus the control group, n = 3.
L144: ((10 mg/Kg body weight/day)
***L147: ** P < 0.05, *** P < 0.01 versus control group; ˄ P < 0.05, ˄˄ P < 0.01, ˄˄˄ P < 0.001 versus the ISO group,-> ** P < 0.01, *** P < 0.001 versus ISO group; ˄ P < 0.05, ˄˄ P < 0.01, ˄˄˄ P < 0.001 versus the control group,
L168: ((10 mg/Kg body weight/day)
L171: The effect of ? on heart tis
***L171: ** P < 0.05, *** P < 0.01 versus control group; ˄ P < 0.05, ˄˄ P < 0.01, ˄˄˄ P < 0.001 versus the ISO group,-> *** P < 0.001 versus ISO group; ˄˄ P < 0.01, ˄˄˄ P < 0.001 versus the control group,
***L183: ** P < 0.05, *** P < 0.01 versus CON group; Ë„ P < 0.05, 183
˄˄ P < 0.01, ˄˄˄ P < 0.001 versus the ISO group ** P < 0.01, *** P < 0.001 versus ISO group; ˄ P < 0.05, ˄˄ P < 0.01 versus the Control group
L201: ((10 mg/Kg body weight/day)
***L204: ** P < 0.05, *** P < 0.01 versus control group; ˄ P < 0.05, ˄˄ P < 0.01, ˄˄˄ P < 0.001 versus the ISO group->*** P < 0.001 versus ISO group; ˄˄ P < 0.01, ˄˄˄ P < 0.001 versus the control group,
L222: ((10 mg/Kg body weight/day)
***L223: (a) The protein expressions of ADH7 and ALDH1A1 in SIRC1 cell lines 223 assessed using Western blotting analysis;
***L266: ** P < 0.05, *** P < 0.01 versus control group; ˄ P < 0.05, ˄˄ P < 0.01, ˄˄˄ P < 0.001 versus the ISO group, => * P < 0.05,** P < 0.01, *** P < 0.001 versus ISO group; ˄˄ P < 0.01, ˄˄˄ P < 0.001 versus the control group,
L232: (Figure 7C and D) -> (Figure 7c and d)
L235: (Figure. 7 B) -> (Figure. 7 b)
L237: (Figure. 7C).-> (Figure. 7c).
L245: ((10 mg/Kg body weight/day)
***L254: ** P < 0.05, *** P < 0.01 versus control group; ˄ P < 0.05, ˄˄ P < 0.01, ˄˄˄ P < 0.001 versus the ISO group,-> ** P < 0.01 versus ISO group; ˄ P < 0.05, ˄˄ P < 0.01, ˄˄˄ P < 0.001 versus the control group,
*L322: ISO-induced myocardial ?? [29].
**L325: However, in comparison with atorvastatin, effect was near to normal.??
**L327: Western blotting analysis revealed a strong mRNA?? expression of ADH7 and ALDH1A1
**L329: increased mRNA?? expression
L342: cardio toxic mice
L348: by previous reports cardioprotective effect of rutin
**L371: In the current study, cleaved caspase-3 and caspase-9?? protein expression levels were higher in ISO-induced MI -> no caspase-9 protein expression data
**L371: caspase-3, caspase-9,-> no caspase-9
L390: Du et al., recently documented
**L392: demonstrated histopathological potential?? [48]
L413: thereby mediate?? the inflammation induced by oxidative stress [55]
L437: s? for LDH
L443: 4.2. Rabbit corneal Cell Culture
L445: 25 cm2 -> 25 cm2 (2 superscript)
L461: 4.4. Neutral Red (NR) Uptake Assay -> 4.4. Neutral red (NR) uptake assay
L474: 4.5. Nitric Oxide (NO) Measurement-> 4.5. Nitric oxide (NO) measurement
*L508: 15 min vs. L534: 15-20 minutes -> min vs. minutes
*L516: cTnTwas measured
**L540: were recorder?
*L543: in mL> in ml
**L646: References: No page number as shown in the following:
L724: Beyoglu Eye J. 2021, doi:10.14744/bej.2021.28190
[Beyoglu Eye J. 2021 Dec 17;6(4):280-284. doi: 10.14744/bej.2021.28190. eCollection 2021.]
L726: Front. Nutr. 2022, 9, doi:10.3389/fnut.2022.914457.
[Front Nutr. 2022 Jul 18:9:914457. doi: 10.3389/fnut.2022.914457. eCollection 2022.]
L765: Front. Pharmacol. 2021, 12,
[Front Pharmacol. 2021 May 19:12:669679. doi: 10.3389/fphar.2021.669679. eCollection 2021.]
L809: In;??
L812: Front. Cardiovasc. Med. 2021, 8,
[Front Cardiovasc Med. 2021 Sep 16:8:740839. doi: 10.3389/fcvm.2021.740839. eCollection 2021.] R60
Comments on the Quality of English LanguageThe English could be improved to more clearly express the research.
Round 2
Reviewer 4 Report (New Reviewer)
Comments and Suggestions for Authors
The present revised manuscript can be accepted for publication after checking the following points:
*L123: ** P < 0.05, and *** P < 0.01 versus ISO group; ˄ P < 0.05, ˄˄ P < 0.01, and ˄˄˄ P < 0.001 versus the control group, n = 3. -> delete *** P < 0.01 and ˄˄˄ P < 0.001, because no such markers in the Fig. 1.
*L129: The CO treatment with-> The pre-treatment with
*L151: ** P < 0.05, and *** P < 0.01 versus ISO group; ˄ P < 0.05, ˄˄ P < 0.01, and ˄˄˄ P < 0.001 versus the control group, n = 3.--> delete *** P < 0.01 and ˄˄˄ P < 0.001, because no such markers in the Fig. 2; ** P < 0.05? -> ** P < 0.01.
*L199: ** P < 0.05, and *** P < 0.01 versus ISO group; ˄ P < 0.05, ˄˄ P < 0.01, and ˄˄˄ P < 0.001 versus the control group, n = 5 -> delete ** P < 0.05 and ˄ P < 0.05, because no such markers in the Fig. 4.
*L212: ** P < 0.05, and *** P < 0.01 versus ISO group; ˄ P < 0.05, ˄˄ P < 0.01, and ˄˄˄ P < 0.001 versus the control group -> delete ˄˄˄ P < 0.001, because no such markers in the Table 1; ** P < 0.05, and *** P < 0.01 ?-> ** P < 0.01, and *** P < 0.001
*L235: ** P < 0.05, and *** P < 0.01 versus ISO group; ˄ P < 0.05, ˄˄ P < 0.01, and ˄˄˄ P < 0.001 versus the control group-> delete ** P < 0.05 and ˄ P < 0.05, because no such markers in the Fig. 5.
*L260: ** P < 0.05, and *** P < 0.01 versus ISO group; Ë„ P < 0.05, ˄˄ P < 0.01, and ˄˄˄ P < 0.001 versus the control group, à** P < 0.05, and *** P < 0.01? ->change to * P < 0.05, ** P < 0.01, and *** P < 0.001; delete Ë„ P < 0.05, because no this marker in the Fig. 6.
*L291: ** P < 0.05, and *** P < 0.01 versus ISO group; ˄ P < 0.05, ˄˄ P < 291 0.01, and ˄˄˄ P < 0.001 versus the control group, -> change ** P < 0.05 to ** P < 0.01; delete *** P < 0.01, because no this marker in the Fig 7.
*L309: it is better to change KEAP and Rutin-KEAP complex to Keap and Rutin-Keap complex in Fig. 8.
L810: 6,280-284 -> 6, 280-284
L812: 9:914457, -> 9, 914457,
L828: Volume 15, 1637–1651-> 15, 1637–1651
L849: 12:669679->12, 669679
L893: Quantitation of nitrate and nitrite in extracellular fluids.-> Quantitation of Nitrate and Nitrite in Extracellular Fluids.
L896: 8:740839-> 8, 740839
Author Response
Please see the attachment.

This manuscript is a resubmission of an earlier submission. The following is a list of the peer review reports and author responses from that submission.
Round 1
Reviewer 1 Report
Comments and Suggestions for Authors
I carefully read the manuscript entitled “Rutin activated Nuclear factor erythroid 2-related factor 2 (Nrf2) to attenuate cornea and heart damage in rats.".
Some comments need to be made to improve it.
The comments are as follows:
- It is suggested that the keywords be organized based on the PubMed MeSH terms and rearranged according to the English alphabet.
- In the "introduction" section, the role of "Rutin" in other diseases should be explained.
- Add a section titled “chemicals” in methods.
- Turn all centrifuge “rpm” to “g”.
- It is mandatory to explain the euthanize of animals.
- In Table 3 change “length” to “product size”
- Unify all significancy symbols in tables and figures.
- Please clarify the aim(s) of the present study at the end of the "Introduction" section.
- The manuscript needs some minor English editing.
Comments on the Quality of English Language
- The manuscript needs some minor English editing.
Reviewer 2 Report
Comments and Suggestions for Authors
The authors investigated the preventive effects of rutin on isoproterenol-induced damage of cornea cells and isoproterenol-induced heart damage in mice. The authors demonstrated that rutin induced the molecules in retinoic acid signaling, antioxidants, anti-apoptotic molecules, and prevented isoproterenol-induced myocardial necrosis. The authors also demonstrated the interaction of rutin and KEAP protein leading to inhibit the binding between KEAP1 and Nrf2. These results indicated that rutin is effective to prevent
cornea defect and myocardial infarction. The purpose in this study is obvious. However, there are substantial points that need to be improved in the data analysis and presentation.
[Major points]
-
The title includes “rats” but the authors used mice in this study.
-
The authors described the synergistic effects between rutin and atorvastatin (Page 1, line 30-31 in Abstract). However, such data is not present in this manuscript.
-
In this study, 32 mice are used (8 mice in each group; 4 groups). In the description in this manuscript (n = 8; Page 5, line 129 etc.), it is thought that all hearts from mice are homogenized and used in ELISA and measurement of enzyme activity etc. However, histological images are present in Figure 7. This contradiction cannot be overlooked. The authors should correct the description properly.
-
The content of the result (Page 8, line 185-191) and the legend of Figure 7 are different. I think that the legend of Figure 7 is correct.
[Minor points]
-
The descriptions for microscopy are missing in Figure 1 and Figure 7.
-
Please add a scale bar in Figure 1.
-
Please spell out cTnT, cTnT at the time of first appearance (Page 4, line 120).
-
In Figure 2B, laminin is detected for internal control. However, the phrase “actin fold” is used in Figure 2C. The authors should correct.
-
Please specify the means of “a, b, c” in Table 1.
-
The method for caspase3 activity (Figure 5) is not described in the Materials and Method section.
-
Figure 7: The scale of the figures should be the same. Moreover, Please add the definition of “Pathological index” in Figure 7. Furthermore, the graph for infiltration and inflammation is not distinguished.
-
Is CCK-8 assay the method to measure viable cells? If so, please specify how the authors used MTT assay and CCK-8 assay (Page 14, line 349, 355, 357-363).
-
Please specify the dose and duration of atorvastatin (Page 15, line 412). Everyday?
-
Please specify the volume of the buffer for homogenization (Page 15, line 417-418).
-
NF-κB is not pro-inflammatory cytokine but a transcription factor (Page 16, line 435). Please improve.
There were some parts that I couldn't understand.
Author Response
Please see the attachement
Thank you.
